# Resveratrol Reverts Tolerance and Restores Susceptibility to Chlorhexidine and Benzalkonium in Gram-Negative Bacteria, Gram-Positive Bacteria and Yeasts

**DOI:** 10.3390/antibiotics11070961

**Published:** 2022-07-18

**Authors:** Antonella Migliaccio, Maria Stabile, Maria Bagattini, Maria Triassi, Rita Berisio, Eliana De Gregorio, Raffaele Zarrilli

**Affiliations:** 1Department of Public Health, University of Naples Federico II, Via S. Pansini 5, 80131 Naples, Italy; antonella.migliaccio10@gmail.com (A.M.); marystabile31@gmail.com (M.S.); maria.bagattini@unina.it (M.B.); triassi@unina.it (M.T.); 2Institute of Biostructures and Bioimaging, National Research Council, 80131 Naples, Italy; rita.berisio@cnr.it; 3Department of Molecular Medicine and Medical Biotechnology, University of Naples Federico II, Via S. Pansini 5, 80131 Naples, Italy

**Keywords:** resveratrol, chlorhexidine, benzalkonium, tolerance, Gram-negative bacteria, Gram-positive bacteria, yeasts

## Abstract

The spread of microorganisms causing health-care associated infection (HAI) is contributed to by their intrinsic tolerance to a variety of biocides, used as antiseptics or disinfectants. The natural monomeric stilbenoid resveratrol (RV) is able to modulate the susceptibility to the chlorhexidine digluconate (CHX) biocide in *Acinetobacter baumannii*. In this study, a panel of reference strains and clinical isolates of Gram-negative bacteria, Gram-positive bacteria and yeasts were analyzed for susceptibility to CHX and benzalkonium chloride (BZK) and found to be tolerant to one or both biocides. The carbonyl cyanide m-chlorophenylhydrazine protonophore (CCCP) efflux pump inhibitor reduced the minimum inhibitory concentration (MIC) and minimum bactericidal concentration (MBC) of CHX and BZK in the majority of strains. RV reduced dose-dependently MIC and MBC of CHX and BZK biocides when used as single agents or in combination in all analyzed strains, but not CHX MIC and MBC in *Pseudomonas aeruginosa*, *Candida albicans*, *Klebsiella pneumoniae*, *Stenotrophomonas maltophilia* and *Burkholderia* spp. strains. In conclusion, RV reverts tolerance and restores susceptibility to CHX and BZK in the majority of microorganisms responsible for HAI. These results indicates that the combination of RV, CHX and BZK may represent a useful strategy to maintain susceptibility to biocides in several nosocomial pathogens.

## 1. Introduction

Multi-drug resistant (MDR) bacterial and yeast pathogens have been recognized as a common cause of health care-associated infections. Among the most frightening of the emerging pathogens is a group of six nosocomial pathogens (*Enterococcus faecium*, *Staphylococcus aureus*, *Klebsiella pneumoniae*, *Acinetobacter baumannii*, *Pseudomonas aeruginosa* and *Enterobacter* spp.) named with the acronym ‘ESKAPE’, because they are capable of ‘escaping’ the biocidal action of antibiotics classified as highly important for human medicine [1,2,3,4,5]. The ESKAPE bacteria are a serious health concern, as they increase the frequency of treatment failures and severity of human infections by adapting to altered environmental conditions and by acquiring resistance determinants [1,2,3,4,5]. Moreover, *Stenotrophomonas maltophilia* and *Burkholderia* spp. are emerging pathogens in cystic fibrosis patients [6]. In addition, severe invasive infections caused by *Candida* spp. that are resistant to antifungal drugs have been increasingly described [7].

The persistence of antimicrobial resistance in MDR pathogens is promoted by a co-selection of antimicrobial resistance with a tolerance to several of the biocides used as antiseptics and disinfectants, such as the bisphenol triclosan (TRI), the quaternary ammonium compounds, benzalkonium chloride (BZK), dequalinium chloride (DQ), cetrimide (CT) and the biguanide chlorhexidine (CHX) [8,9,10]. CHX is a microbicidal agent, which is currently used for hand hygiene, skin antisepsis, oral care and patient washing [9]. BZK has been widely used as a disinfectant in hospitals, or as an antiseptic in antimicrobial soaps [10]. Tolerance to CHX and BZK is emerging in major nosocomial pathogens [9,10,11,12,13,14,15,16].

A reduced susceptibility to biocides in *A. baumannii*, *K. pneumoniae* and other pathogens has been correlated with the activation of different efflux pump (EP) systems [13,16,17,18,19,20]. In particular, the inhibition of the AdeB RND superfamily and AceI PACE superfamily EP systems has been demonstrated to restore susceptibility to CHX and BZK in *A. baumannii* [19,20]. In addition, it has been demonstrated that the natural monomeric stilbenoid resveratrol (RV) [21], which has been demonstrated to possess antimicrobial activity against a wide range of bacterial and fungal species [22,23], is able to inhibit EP expression and restore susceptibility to CHX and BZK biocides in *A. baumannii* [19,24].

The objectives of the present study were to: (i) analyze the susceptibility to BZK and CHX biocides in a panel of reference strains and clinical isolates of Gram-negative bacteria, Gram-positive bacteria and yeasts; (ii) analyze whether the natural compound RV at non-toxic concentrations may modulate and restore susceptibility to CHX and BZK in the above pathogens.

## 2. Results and Discussion

### 2.1. Antimicrobial Activity of CHX and BZK against a Panel of Reference Strains and Clinical Isolates of Gram-Negative Bacteria, Gram-Positive Bacteria and Yeasts

The antimicrobial activity of CHX and BZK was assessed by broth microdilution assay against 151 reference strains and clinical isolates of Gram-negative bacteria, Gram-positive bacteria and yeasts, which included the ESKAPE bacteria, *S. maltophilia*, *Burkholderia* spp. and *Candida* spp. (Figure 1 and Figure 2; Appendix A). *A. baumannii*, *K. pneumoniae*, *K. aerogenes*, *P. aeruginosa*, *E. coli* EC-Na1-Na4, *S. maltophilia*, *Burkholderia* spp., *Enterococcus* spp. and *Candida* spp. strains showed CHX minimum inhibitory concentration (MIC), minimum bactericidal concentration (MBC) or minimum fungicidal concentration (MFC), in the case of *Candida* spp., values ranging from 4–64 mg/L and 4–128 mg/L, respectively, and were considered tolerant to CHX (Figure 1; Appendix A). Instead, *E. coli* ATCC 25922, *E. coli* ATCC 35218, *S. aureus* and *S. epidermidis* strains showed CHX MIC and MBC values of 1–2 mg/L and were considered susceptible (Figure 1; Appendix A). The median MIC and MBC values of CHX were significantly higher in *A. baumannii*, *E. cloacae*, *Klebsiella* spp., *S. maltophilia* and *P. aeruginosa* strains compared with those of the susceptible strains (*p* < 0.05, *p* < 0.01 and *p* < 0.001, respectively).

In addition, the *A. baumannii*, *K. pneumoniae*, *K. aerogenes*, *P. aeruginosa*, *E. coli* ATCC 25922, *E. coli* ATCC 35218, *S. maltophilia*, *Burkholderia* spp., *Enterococcus* spp., *S. aureus* ATCC 43300 and *Candida* spp. strains showed both MIC and MBC (MFC in the case of *Candida* spp.) values for BZK within the range of 4–64 mg/L and were considered tolerant (Figure 2; Appendix A). Instead, *E. coli* EC-Na1-Na4, *S. aureus* ATCC 25923, *S. aureus* ATCC 29213, *S. epidermidis* ATCC 12282, *C. krusei* 81667 and *C. tropicalis* 61220 showed BZK MIC and MBC (MFC) values of 1–2 mg/L and were considered susceptible (Figure 2; Appendix A). The median MIC and MBC values of BZK were significantly higher in the *P. aeruginosa* strains compared with those of susceptible strains (*p* < 0.05 and *p* < 0.001, respectively). The above overall data are in agreement with previous studies showing that microbial pathogens responsible for health care-associated infection, in particular Gram-negative bacteria such as *K. pneumoniae* and *P. aeruginosa*, are highly tolerant to CHX and BZK biocides [11,12,13,14,15].

### 2.2. Effect of Carbonyl Cyanide M-Chlorophenylhydrazine Protonophore (CCCP) EP Inhibitor on CHX and BZK MIC and MBC against Gram-Negative Bacteria, Gram-Positive Bacteria and Yeasts

To evaluate whether the tolerance to CHX and BZK was mediated by the activation of EP, as demonstrated in the Gram-negative and Gram-positive bacteria [13,14,15,16,17,18,19], we analyzed the effect of the EP inhibitor CCCP on CHX and BZK susceptibility. As shown in Table 1, CCCP reduced dose-dependently the CHX MIC and MBC or MFC in *A. baumannii*, *Burkholderia* spp., *K. pneumoniae*, *K. aerogenes*, *Enterobacter* spp., *Enterococcus* spp., *S. maltophilia*, *S. enterica* and *C. parapsilosis* with a decrease between 4- and 64-fold. The inhibitory effect of CCCP was less evident in the *P. aeruginosa* and *Candida* spp. strains, in which the MIC and MBC (MCF) of CHX were reduced only by one-fold or not affected (Table 1). In addition, CCCP reduced the MIC and MBC (MFC in the case of *Candida* spp.) of BZK by one- to four-fold in *A. baumannii*, *Candida* spp., *B. gladioli*, *B. dolosa*, *S. enterica* and *S. maltophilia* strains, while it had no effect on the BZK MIC and MBC in other *Burkholderia* species, *K. pneumoniae* and *P. aeruginosa* strains (Table 1). In accordance with the previous data [19], the reduction in MIC and MBC of CHX and BZK due to CCCP was only four- and two-fold, respectively, in *A. baumannii* ATCC 19606 carrying the deletion of the adeB EP gene, consistent with the role of AdeB in CHX and BZK extrusion (Table 1). The data shown herein indicate that tolerance to CHX and BZK is mediated by activation of the EPs and are in agreement with previous publications showing that CHX and BZK tolerance in *K. pneumoniae* and *A. baumannii* clinical isolates is mediated by RND superfamily EP activation [13,16], and that CHX tolerance in *P. aeruginosa* clinical isolates is mediated by an increased expression of the *mexA, mexC, mexE* and *mexX* EP genes, and a decreased expression of the *oprD* gene [15].

### 2.3. Effect of RV on CHX and BZK MIC and MBC (or MFC) against Gram-Negative Bacteria, Gram-Positive Bacteria and Yeasts

We next analyzed if the natural monomeric stilbenoid RV [21], which was demonstrated to regulate EPs expression and counteract the tolerance to CHX and BZK in *A. baumannii* [19,24], may restore susceptibility in the Gram-negative bacteria, Gram-positive bacteria and yeasts included in the study. Our previous data demonstrated that RV at >1024 mg/L has no antimicrobial activity against *A. baumannii* ATCC19606 [19]. In agreement with this, the RV showed no antimicrobial activity against all of the Gram-negative bacteria, Gram-positive bacteria and yeasts included in the study with MIC values > 1024 mg/L (Appendix A). On the other hand, the data shown herein are partly in agreement with previous studies showing that RV at high concentrations has antimicrobial activity against *S. aureus*, *E. faecalis*, *E. faecium*, *E. coli* and *Candida* spp. strains [22,23]. The discrepancies between our data and previous studies [22,23] may depend on different strains and/or different experimental conditions.

We then analyze the effect of RV in combination with CHX or BZK. The objectives of our experiments were to identify which RV concentrations were able to revert tolerance and restore susceptibility to the CHX and BZK biocides.

As shown in Table 2, RV from 32 to 256 mg/L decreased dose-dependently the MIC and MBC (MCF for *Candida* spp.) of CHX in 33 selected strains among the Gram-negative bacteria, Gram-positive bacteria and yeasts and restored CHX susceptibility in most of the strains, but not in *K. pneumoniae* ATCC 700603, all of the *P. aeruginosa* or *Candida* spp. strains. Interestingly, a positive correlation was found between the RV effect on CHX MIC and the MBC and CCCP effect on CHX MIC and MBC, *P. aeruginosa* and *Candida* spp. strains showing high CHX MIC and MBC values after treatment with RV or CCCP (Table 1 and Table 2; Appendix A) (r = 0.893, *p* < 0.001). The above data indicate that the RV effect on the inhibition of CHX tolerance in the Gram negative-bacteria, Gram-positive bacteria and yeasts is mediated by the inhibition of EP activity.

Furthermore, increasing the doses of RV up to 128 mg/L decreased dose-dependently the BZK MIC and MBC (MCF for *Candida* spp.) and restored the BZK susceptibility in most of the strains, but not *B. cenocepacia* LMG16654, *B. dolosa* LMG21443, *B. multivorans* LMG16654, *E. cloacae* ATCC 13047, *K. pneumoniae* ATCC 700603, *S. maltophilia* K279 or all of the *P. aeruginosa* strains (Table 3). A positive correlation was also found between the RV effect on BZK MIC and the MBC and CCCP effect on BZK MIC and MBC (Table 1 and Table 3; Appendix A) (r = 0.775, *p* < 0.01), thus suggesting that the inhibition of the EPs activation might be involved in the inhibitory effect of RV on tolerance to BZK. This is in agreement with a previous publication that showed that RV inhibited basal and CHX-induced expression of the AdeB RND superfamily and the AceI PACE superfamily EP systems in *A. baumannii* [19].

### 2.4. Effect of RV on CHX and BZK Combination against Gram-Negative Bacteria, Gram-Positive Bacteria and Yeasts

Because the CHX and BZK biocides/antiseptics are currently used in combination [25,26,27], we analyzed the effect of RV on the susceptibility of Gram-negative bacteria, Gram-positive bacteria and yeasts to the CHX and BZK combined treatment. As shown in Table 4, the CHX and BZK combination inhibited the CHX or BZK MIC in 14 out 21 strains, showing either a synergistic or additive effect in 14 and 5 strains, respectively, but an indifferent effect was observed in the *C. albicans* 62033 and *P. aeruginosa* PAO1. The CHX and BZK combination in the presence of 32 mg/L RV inhibited the CHX and BZK MIC and MBC in all of the strains, and restored the CHX or BZK susceptibility in 16 out of 21 strains, resulting in a synergistic or additive effect in 18 and 3 strains, respectively (Table 4). Moreover, the CHX and BZK combination in the presence of RV at 64 mg/L restored the CHX or BZK susceptibility in all of the strains, and showed a synergistic or additive effect in 21 strains (Table 4). In particular, the CHX and BZK combination in the presence of 64 mg/L RV restored the BZK susceptibility in all of the strains, while the combination did not affect the CHX tolerance in three *Burkholderia* spp. strains, *C. albicans* 62033, *S. maltophilia* k279, *K. pneumoniae* ATCC 700603 and all four of the *P. aeruginosa* strains were still tolerant to CHX (Table 4). The reduced ability of RV to restore CHX susceptibility compared to BZK susceptibility may be dependent on the elevated EP activation, which was demonstrated to regulate tolerance to CHX in the *K. pneumoniae* [13] and *P. aeruginosa* [15] strains. Future experiments are necessary to validate this hypothesis.

## 3. Materials and Methods

### 3.1. Bacterial Strain, Growth Condition and Reagents

A collection of 132 Gram-negative bacteria, 9 Gram-positive bacteria and 10 *Candida* spp. strains was analyzed in the study (Appendix A). The collection included either the reference strains, which were identified with their ATCC number, or clinical isolates, which were identified with their original number (Appendix A). The origin and characteristics of all of the strains were described in the references listed in Appendix A. No ethical approval was required for the study because there was no access to patients’ data. The reference and clinical strains were cultured under aerobic conditions in standard selective media at 37 °C, but the *S. maltophilia* LMG 10991, LMG 10853 and LMG 10871 strains were grown at 30 °C. The chemical reagents, chlorhexidine digluconate (CHX), benzalkonium chloride (alkylbenzyldimethylammonium chloride (BZK), carbonyl cyanide m-chlorophenylhydrazine (CCCP) and resveratrol (3,5,4′-trihydroxy-trans-stilbene, RV), were purchased from Sigma-Aldrich (Sigma, Milan, Italy).

### 3.2. Determination of Minimum Inhibitory Concentration and Minimum Bactericidal Concentration

The CCCP and RV were dissolved in dimethyl sulfoxide (DMSO), while the CHX and BZK were dissolved in H_2_O. Two-fold serial dilutions of CHX and BZK, (0.06–1024 mg/L), RV (32–256 mg/L) or CCCP (1–2 and 4 mg/L), were prepared in triplicate and placed into a polystyrene 96-well plate. The bacterial suspensions were prepared by growing overnight in nutrient media with agar and adjusting the turbidity to 0.5 McFarland using a BD PhoenixSpec™ nephelometer. Subsequently, the bacterial cells were diluted in cation-adjusted Mueller–Hinton broth (CAMHB) to a final culture density of approximately 1 × 10^6^ CFU/mL. Only the CAMHB was added into the negative control wells, and wells with no compounds were used on each plate as the positive growth control. The plates were incubated at 30 °C or 37 °C for 18–24 h. The MIC and MBC of CHX, BZK, CCCP or RV were determined by a manual microdilution method, according to the recommended procedures by the European Committee for Antimicrobial Susceptibility Testing (Eucast) of the European Society of Clinical Microbiology and Infectious Diseases (Escmid) [28], and the Clinical and Laboratory Standards (CLSI) [29]. On the other hand, the yeast suspensions were prepared by growing overnight in Sabouraud dextrose agar plates, and adjusting the turbidity to 0.5 McFarland, using a BD PhoenixSpec™ nephelometer. Then, the 5 × 10^5^ CFU/mL yeasts were inoculated in RPMI buffered with morpholinepropane sulfonic acid (MOPS) (pH. 7) containing glucose 2%. The non-treated yeasts were used as the positive controls. The MIC of CHX, BZK, CCCP or RV were determined by a manual microdilution method, according to the recommended procedures by the Subcommittee on Antifungal Susceptibility Testing (AFST) European Committee for Antimicrobial Susceptibility Testing (EUCAST) of the ESCMID [30]. Finally, the plates were incubated at 37 °C for 18–24 h. The susceptibility was assessed to the MIC value < 4 mg/L, as described for *A. baumannii* [16]. The strains showing MIC values < 4 mg/L were considered susceptible, while the strains having MIC values ≥ 4 mg/L were considered tolerant. In order to evaluate the minimum bactericidal or fungicide concentration, 20 µL of bacteria or yeast suspensions from wells without visible growth were transferred to the respective plates. These plates were incubated at 30 °C or 37 °C and checked for growth after 24 h. All of the tests were performed in triplicate and repeated three times.

### 3.3. In Vitro Combination Studies

The tests were carried out using the checkerboard method, according to the previously reported method [31]. The serial dilutions of CHX (0.06–164 mg/L) or BZK (0.06–164 mg/L) were prepared and combined with fixed concentrations of resveratrol (32–256 mg/L) or CCCP (1, 2 and 4 mg/L). Subsequently, 1 × 10^6^ CFU/mL bacterial cells in CAMHB and 5 × 10^5^ CFU/mL yeasts in RPMI-MOPS were added to each well of the microtiter plate. Then, the plates were incubated at 30 °C or 37 °C for 18–24 h. Furthermore, the checkerboard method was used to evaluate the MICs for the combination of CHX and BZK from 0.06 to 64 mg/L with RV at fixed concentrations of 32 or 64 mg/L. Afterward, the microtiter was incubated with 1 × 10^6^ CFU/mL bacterial cells in CAMHB or 5 × 10^5^ CFU/mL yeasts in RPMI-MOPS. The plates were then incubated at 37 °C for 18–24 h. The combined effects were then determined by calculating the fractional inhibitory concentration (FIC) index as follows: FICI = FIC_A_ + FIC_B_, where FIC_A_ is the ratio of the MIC of CHX and BZK with RV (32 or 64 mg/L) combination and the MIC of CHX with RV (32 or 64 mg/L) alone, and FIC_B_ is the ratio of the MIC CHX and BZK with RV (32 or 64 mg/L) combination and the MIC of BZK with RV (32 or 64 mg/L) alone. The FIC index was interpreted as synergy (FICI ≤ 0.5), additive (FICI > 0.5 to ≤1.0), indifference (FICI > 1.0 to ≤2.0) and antagonism (FICI > 2.0). All of the experiments were repeated three times [32].

### 3.4. Statistical Analysis

All of the statistical analyses were performed with GraphPad Prism 8 software (GraphPad, San Diego, CA, USA). The correlations were evaluated by regression analysis, using the Pearson’s correlation coefficient (r). All of the results are presented as arithmetic means ± standard deviations. The significance of the differences was evaluated using one-way ANOVA, followed by Bonferroni’s comparison post-hoc tests. The differences were considered statistically significant if *p* < 0.05.

## 4. Conclusions

The tolerance of dangerous Gram-negative bacteria, Gram-positive bacteria and yeasts to commonly used biocides, such as CHX and BZK, is becoming a serious nosocomial problem [6,7,8,11,12,16,17].

Although chemically different, CHX (a cationic poly-biguanide) and BZK (a quaternary ammonium compound) share a cationic nature, that makes them able to bind to the negatively-charged sites on the cell wall; thus, destabilizing it and interfering with osmosis [4,5,6]. However, the bacteria have developed mechanisms to resist the attack of biocides, e.g., extruding them through EPs, resulting in the clinically observed biocide tolerance [9,10,11,12,13,16,17]. This phenomenon has prompted us to search for effective formulations, able to exert their antimicrobial action on the resistant bacterial strains. We have demonstrated synergy in the bactericidal effect of CHX and BZK in a large panel of Gram-negative and Gram-positive bacteria, the highest effects being observed for *A. baumannii* ACICU, *B. dolosa* LMG 21443, *B. multivorans* LMG 16665, *B. cenocepacia* LMG 16654, *P. aeruginosa* RP73, *P. aeruginosa* PA14, *K. pneumoniae* ATCC 700603, *K. pneumoniae* kp-Mo-7, *S. enterica* ATCC 13076 and *S. aureus* ATCC 43300.

Importantly, a synergistic microbicidal effect was observed when the two biocides were combined with resveratrol, which we previously proved affected the expression level of the EPs [11]. This finding has a strong applicative potential for the preparation of disinfectant/antiseptic formulations containing the three components, to be used against strongly tolerant Gram-negative bacteria, Gram-positive bacteria and yeasts.

## Figures and Tables

**Figure 1 antibiotics-11-00961-f001:**
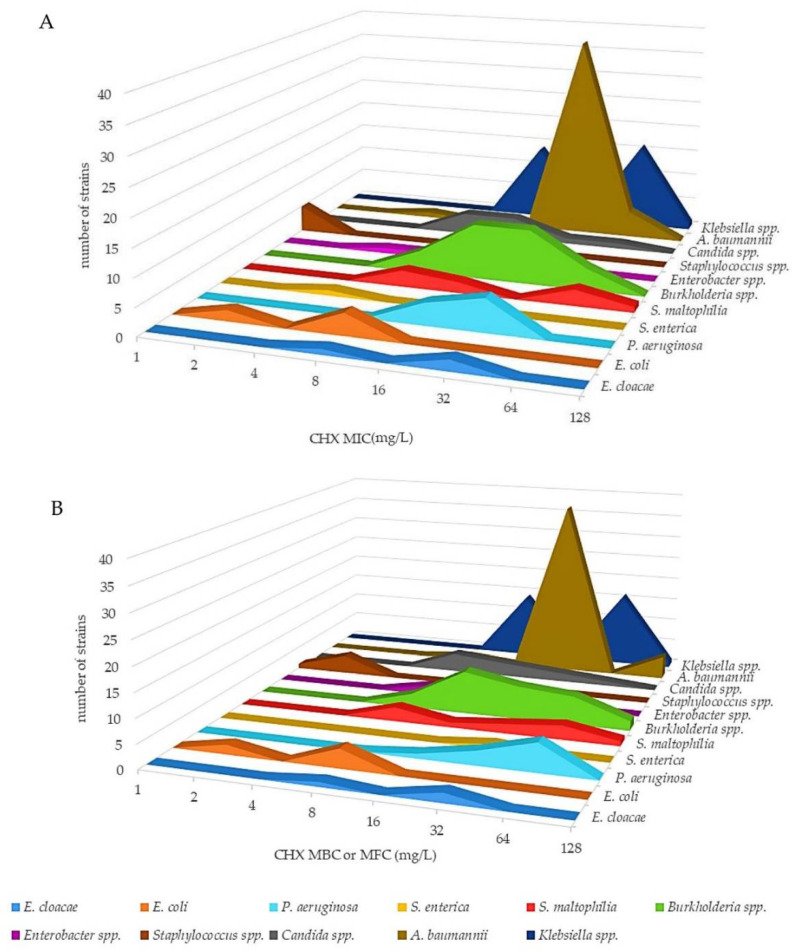
Three-dimensional aerogram of CHX MICs (mg/L) (**A**) and MBCs (MFCs in the case of *Candida*) (mg/L) (**B**) of reference strains and clinical isolates of Gram-negative bacteria, Gram-positive bacteria and yeasts.

**Figure 2 antibiotics-11-00961-f002:**
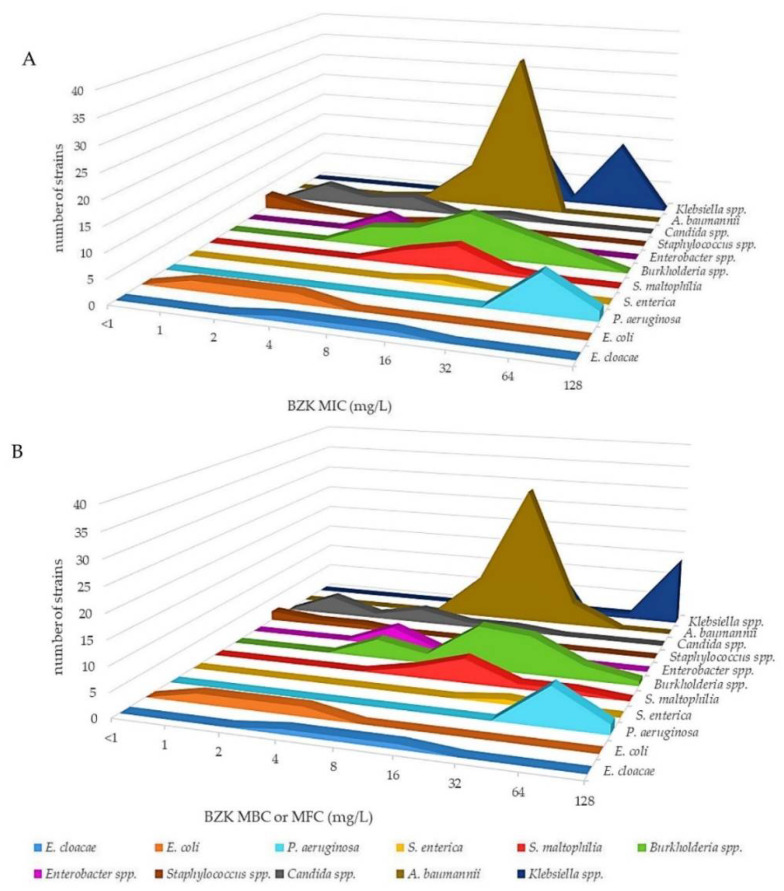
Three-dimensional aerogram of BZK MICs (mg/L) (**A**) and MBCs (MFCs in the case of *Candida*) (mg/L) (**B**) of reference strains and clinical isolates of Gram-negative bacteria, Gram-positive bacteria and yeasts.

**Table 1 antibiotics-11-00961-t001:** Effect of CCCP (mg/L) efflux pump inhibitor on CHX and BZK MIC (mg/L) and MBC (MFC in the case of *Candida* spp.) (mg/L) against Gram-negative bacteria, Gram-positive bacteria and yeasts.

Strain	CHX MIC (MBC or MFC)				BZK MIC (MBC or MFC)			
	CCCP				CCCP			
	0	1	2	4	0	1	2	4
*A. baumannii* ATCC 19606	32 (32)	8 (8)	4 (4)	1 (1)	16 (32)	8 (16)	8 (16)	8 (16)
*A. baumannii *∆*adeB* ATCC 19606	4 (4)	2 (2)	2 (2)	2 (2)	8 (8)	4 (4)	4 (4)	2 (2)
*A. baumannii* ACICU	64 (128)	32 (64)	8 (16)	8 (16)	16 (16)	16 (16)	16 (16)	16 (16)
*A. baumannii* AYE	32 (32)	8 (8)	4 (4)	1 (1)	16 (32)	8 (16)	8 (16)	8 (16)
*B. cenocepacia* LMG 16654	64 (64)	64 (64)	32 (32)	16 (16)	64 (128)	64 (128)	64 (128)	64 (128)
*B.cepacia* LMG 2161	32 (64)	32 (32)	8 (8)	4 (4)	32 (32)	32 (32)	16 (32)	16 (32)
*B. vietnamiensis* LMG 22486	32 (32)	32 (32)	8 (8)	4 (4)	32 (32)	32 (32)	16 (32)	16 (32)
*B. gladioli* LMG 2121	16 (16)	16 (32)	8 (16)	4 (4)	4 (4)	4 (4)	1 (1)	1 (1)
*B. dolosa* LMG 21443	16 (32)	16 (32)	8 (16)	4 (4)	8 (16)	8 (16)	4 (8)	4 (8)
*B. multivorans* LMG 16665	64 (64)	32 (32)	16 (16)	8 (8)	32 (64)	32 (32)	16 (16)	8 (8)
*E. cloacae* ATCC13047	8 (8)	4 (8)	2 (4)	1 (2)	32 (64)	32 (64)	32 (64)	32 (64)
*E. cloacae* EnC-Na-1	32 (32)	16 (16)	4 (8)	2 (4)	4 (4)	4 (8)	4 (8)	4 (8)
*K. aerogenes* ATCC 13048	32 (32)	8 (16)	4 (4)	2 (2)	32 (32)	16 (16)	16 (16)	16 (16)
*K. pneumoniae* ATCC 700603	128 (128)	32 (32)	8 (8)	4 (4)	32 (64)	32 (64)	32 (64)	32 (64)
*K. pneumoniae* KP-Mo-7	64 (64)	16 (16)	4 (4)	1 (1)	32 (32)	32 (32)	32 (32)	32 (32)
*K. pneumoniae* KP-Mo-6	64 (64)	16 (16)	4 (4)	1 (1)	32 (32)	32 (32)	32 (32)	32 (32)
*P. aeruginosa* RP73	32 (32)	32 (32)	16 (16)	16 (16)	64 (128)	64 (64)	64 (64)	64 (64)
*P. aeruginosa* PAO1	16 (16)	16 (16)	8 (8)	8 (8)	64 (128)	64 (128)	64 (128)	64 (64)
*P. aeruginosa* PA14	16 (32)	16 (16)	8 (8)	8 (8)	64 (64)	64 (64)	64 (64)	64 (64)
*P. aeruginosa* PA-Na-1	32 (64)	32 (32)	16 (16)	16 (16)	64 (128)	64 (64)	64 (64)	64 (64)
*S. enterica* ATCC 13076	4 (4)	1 (1)	0.5 (1)	0.5 (1)	16 (16)	16 (16)	16 (16)	4 (4)
*S. maltophilia* K279	32 (32)	8 (8)	2 (2)	2 (2)	16 (16)	16 (16)	16 (16)	16 (16)
*S. maltophilia* LMG 10853	8 (8)	4 (4)	2 (2)	0.5 (1)	8 (8)	4 (8)	4 (8)	4 (8)
*S. maltophilia* OBGTC20	64 (64)	32 (32)	8 (8)	2 (2)	32 (32)	16 (16)	16 (16)	16 (16)
*E. faecalis* ATCC 29212	64 (64)	16 (16)	16 (16)	8 (8)	4 (8)	4 (8)	4 (8)	4 (8)
*E. faecium* ATCC 6057	8 (8)	4 (4)	2 (2)	0.5 (1)	4 (8)	4 (8)	4 (8)	2 (4)
*S. aureus* ATCC 43300	1 (2)	<1 (<1)	<1 (<1)	<1 (<1)	8 (8)	<1 (<1)	<1 (<1)	<1 (<1)
*C. albicans* ATCC 10231	16 (16)	16 (16)	16 (16)	8 (8)	2 (2)	<1 (<1)	<1 (<1)	<1 (<1)
*C. albicans* 62033	16 (16)	16 (16)	16 (16)	8 (8)	4 (4)	4 (4)	1 (1)	1 (1)
*C. parapsilosis* 4609	32 (32)	16 (16)	16 (16)	4 (4)	2 (2)	<1 (<1)	<1 (<1)	<1 (<1)
*C. krusei* 81667	8 (8)	8 (8)	8 (8)	8 (8)	1 (1)	<1 (<1)	<1 (<1)	<1 (<1)
*C. glabrata* 61112	16 (16)	16 (16)	16 (16)	8 (8)	2 (2)	<1 (<1)	<1 (<1)	<1 (<1)
*C. tropicalis* 61220	8 (16)	8 (8)	8 (8)	8 (8)	1 (1)	<1 (<1)	<1 (<1)	<1 (<1)

The numbers outside and within the brackets indicate MIC and MBC or MCF values, respectively, and are expressed in mg/L. The numbers 0, 1, 2, and 4 indicate CCCP concentrations and are expressed in mg/L.

**Table 2 antibiotics-11-00961-t002:** Effect of RV (mg/L) on CHX MIC (mg/L) and MBC (MFC in the case of *Candida* spp.) (mg/L) against Gram-negative bacteria, Gram-positive bacteria and yeasts.

Strain	CHX MIC (MBC or MFC)				
	RV				
	0	32	64	128	256
*A. baumannii* ATCC 19606	32 (32)	8 (16)	4 (8)	0.5 (2)	0.125 (0.125)
*A. baumannii *∆*adeB* ATCC 19606	4 (4)	4 (4)	4 (4)	0.5 (0.5)	0.125 (0.125)
*A. baumannii* ACICU	64 (128)	16 (32)	4 (8)	0.5 (0.5)	0.5 (0.5)
*A. baumannii* AYE	32 (32)	4 (8)	2 (4)	0.5 (2)	0.125 (0.125)
*B. cenocepacia* LMG 16654	64 (64)	32 (32)	32 (32)	4 (4)	2 (2)
*B. cepacia* LMG 2161	32 (32)	16 (16)	16 (16)	2 (4)	2 (2)
*B. vietnamiensis* LMG 22486	32 (32)	32 (32)	16 (16)	2 (2)	0.5 (0.5)
*B. gladioli* LMG 2121	16 (16)	4 (4)	4 (4)	4 (4)	1 (1)
*B. dolosa* LMG 21443	16 (16)	8 (8)	8 (4)	4 (4)	2 (2)
*B. multivorans* LMG 16665	64 (64)	64 (64)	32 (32)	4 (8)	2 (2)
*E. cloacae* ATCC 13047	8 (8)	4 (8)	2 (2)	0.5 (1)	0.5 (0.5)
*E. cloacae* EnC-Na-1	32 (32)	8 (8)	4 (8)	4 (4)	2 (2)
*K. aerogenes* ATCC 13048	32 (32)	8 (16)	8 (16)	2 (2)	1 (1)
*K. pneumoniae* ATCC 700603	128 (128)	64 (64)	32 (32)	8 (8)	4 (4)
*K. pneumoniae* KP-Mo-7	64 (64)	32 (32)	32 (32)	4 (8)	2 (4)
*K. pneumoniae* KP-Mo-6	64 (64)	16 (32)	16 (16)	8 (16)	2 (2)
*P. aeruginosa* RP73	32 (32)	8 (16)	8 (16)	8 (16)	4 (8)
*P. aeruginosa* PAO1	16 (16)	4 (8)	4 (8)	4 (4)	4 (4)
*P. aeruginosa* PA14	16 (32)	16 (32)	8 (32)	4 (8)	4 (8)
*P. aeruginosa* PA-Na-1	32 (64)	8 (16)	4 (16)	4 (8)	4 (8)
*S. enterica* ATCC 13076	4 (4)	1 (1)	0.5 (1)	0.5 (1)	0.5 (1)
*S. maltophilia* K279	64 (128)	32 (32)	16 (16)	4 (8)	0.5 (1)
*S. maltophilia* LMG 10853	8 (8)	2 (2)	0.5 (1)	0.5 (1)	0.5 (0.5)
*S. maltophilia* OBGTC20	64 (64)	16 (16)	16 (16)	4 (4)	1 (1)
*E. faecalis* ATCC 29212	32 (32)	16 (16)	4 (8)	2 (2)	0.5 (0.5)
*E. faecium* ATCC 6057	8 (8)	8 (8)	4 (4)	2 (2)	0.5 (0.5)
*S. aureus* ATCC 43300	1 (2)	0.25 (0.25)	0.25 (0.25)	0.125 (0.25)	0.125 (0.25)
*C. albicans* ATCC 10231	16 (32)	16 (32)	16 (16)	8 (8)	8 (8)
*C. albicans* 62033	16 (16)	16 (16)	16 (16)	16 (16)	8 (8)
*C. parapsilosis* 4609	64 (64)	32 (64)	16 (32)	16 (32)	8 (8)
*C. krusei* 81667	8 (8)	4 (8)	4 (4)	2 (4)	2 (4)
*C. glabrata* 61112	16 (16)	8 (16)	8 (16)	8 (8)	8 (8)
*C. tropicalis* 61220	8 (16)	8 (8)	8 (8)	4 (4)	4 (4)

The numbers outside and within the brackets indicate MIC and MBC or MCF values, respectively, and are expressed in mg/L. The numbers 0, 32, 64, 128 and 256 indicate RV concentrations and are expressed in mg/L.

**Table 3 antibiotics-11-00961-t003:** Effect of RV (mg/L) on BZK MIC (mg/L) and MBC (MFC in the case of *Candida* spp.) (mg/L) against Gram-negative bacteria, Gram-positive bacteria and yeasts.

Strain	BZK MIC (MBC or MFC)			
	RV			
	0	32	64	128
*A. baumannii* ATCC 19606	16 (32)	8 (16)	4 (16)	0.5 (1)
*A. baumannii *∆*adeB* ATCC 19606	8 (8)	2 (4)	1 (1)	0.25 (0.5)
*A. baumannii* ACICU	16 (32)	2 (4)	1 (2)	0.5 (1)
*A. baumannii* AYE	8 (8)	1 (2)	1 (1)	0.25 (1)
*B. cenocepacia* LMG 16654	64 (64)	32 (32)	32 (32)	4 (4)
*B.cepacia* LMG 2161	32 (32)	16 (16)	16 (16)	2 (4)
*B. vietnamiensis* LMG 22486	32 (32)	32 (32)	16 (16)	2 (2)
*B. gladioli* LMG2121	4 (4)	2 (2)	0.5 (1)	0.5 (1)
*B. dolosa* LMG21443	32 (32)	16 (16)	16 (16)	8 (8)
*B. multivorans* LMG 16665	64 (64)	64 (64)	32 (32)	4 (4)
*E. cloacae* ATCC 13047	32 (32)	16 (32)	16 (16)	16 (16)
*E. cloacae* EnC-Na-1	4 (4)	4 (4)	2 (2)	2 (2)
*K. aerogenes* ATCC 13048	8 (8)	8 (8)	4 (4)	2 (2)
*K. pneumoniae* ATCC 700603	32 (32)	32 (32)	32 (32)	16 (16)
*K. pneumoniae* KP-Mo-7	16 (16)	8 (16)	4 (8)	2 (2)
*K. pneumoniae* KP-Mo-6	16 (16)	16 (16)	4 (8)	2 (2)
*P. aeruginosa* RP73	32 (32)	32 (32)	32 (32)	16 (16)
*P. aeruginosa* PAO1	64 (64)	64 (64)	64 (64)	32 (32)
*P. aeruginosa* PA14	64 (64)	32 (64)	32 (64)	32 (64)
*P. aeruginosa* PA-Na-1	64 (128)	64 (128)	64 (64)	64 (64)
*S. enterica* ATCC 13076	32 (32)	32 (32)	16 (16)	8 (8)
*S. maltophilia* K279	16 (16)	16 (16)	8 (16)	4 (8)
*S. maltophilia* LMG 10853	8 (8)	1 (1)	0.5 (1)	0.5 (1)
*S. maltophilia* OBGTC20	32 (32)	8 (16)	4 (8)	2 (2)
*E. faecalis* ATCC 29212	4 (8)	4 (8)	4 (8)	2 (2)
*E. faecium* ATCC 6057	4 (4)	4 (4)	1 (1)	0.5 (1)
*S. aureus* ATCC 43300	8 (16)	0.5 (0.5)	0.25 (0.25)	0.125 (0.25)
*C. albicans* ATCC 10231	4 (4)	4 (4)	2 (2)	2 (2)
*C. albicans* 62033	4 (4)	4 (4)	2 (2)	2 (2)
*C. parapsilosis* 4609	4 (8)	4 (4)	2 (2)	2 (2)
*C. krusei* 81667	1 (1)	0.5 (1)	0.5 (1)	0.5 (1)
*C. glabrata* 61112	2 (2)	0.5 (1)	0.5 (1)	0.5 (1)
*C. tropicalis* 61220	1 (1)	0.5 (1)	0.125 (0.125)	0.125 (0.125)

The numbers outside and within the brackets indicate MIC and MBC or MCF values, respectively, and are expressed in mg/L. The numbers 0, 32, 64 and 128 indicate RV concentrations and are expressed in mg/L.

**Table 4 antibiotics-11-00961-t004:** Effect of RV (mg/L) on CHX and BZK MIC (mg/L) and MBC (MFC in the case of *Candida* spp.) (mg/L) in combination against Gram-negative bacteria, Gram-positive bacteria and yeasts.

Strain	CHX MIC (MBC or MFC)	BZK MIC (MBC or MFC)	0 RV		FIC * Index ^(a)^	32 RV		FIC * Index ^(b)^	64 RV		FIC * Index ^(c)^
			CHX + BZK MIC (MBC or MFC)			CHX + BZK MIC (MBC or MFC)			CHX + BZK MIC (MBC or MFC)		
*A. baumannii* ATCC 19606	32 (32)	16 (16)	8 (8)	4 (4)	0.5	2 (2)	2 (2)	0.187	2 (2)	0.5 (0.5)	0.093
*A. baumannii* ∆*adeB* ATCC 19606	4 (4)	8 (8)	1 (1)	2 (2)	0.5	2 (2)	0.06 (0.06)	0.50	0.5 (0.5)	0.5 (0.5)	0.18
*A. baumannii* ACICU	64 (128)	16 (32)	8 (8)	2 (2)	0.25	2 (2)	0.5 (0.5)	0.062	2 (2)	0.5 (0.5)	0.062
*B. cenocepacia* LMG 16654	64 (64)	64 (64)	8 (8)	2 (2)	0.15	8 (8)	2 (2)	0.15	8 (8)	0.5 (0.5)	0.13
*B. dolosa* LMG 21443	16 (16)	8 (8)	4 (8)	2 (2)	0.31	4 (4)	1 (1)	0.28	4 (4)	0.5 (0.5)	0.26
*B. multivorans* LMG16665	64 (64)	32 (64)	8 (8)	8 (8)	0.25	8 (8)	8 (8)	0.25	8 (8)	0.5 (0.5)	0.13
*E. cloacae* ATCC 13047	8 (8)	8 (8)	2 (2)	4 (4)	0.5	2 (2)	4 (4)	0.5	2 (2)	2 (2)	0.375
*K. pneumoniae* ATCC 700603	128 (128)	16 (16)	16 (32)	4 (4)	0.25	4 (4)	2 (2)	0.093	4 (4)	2 (2)	0.093
*K. pneumoniae* kp-Mo-7	64 (64)	16 (16)	2 (2)	2 (2)	0.156	2 (2)	1 (1)	0.092	1 (1)	0.5 (0.5)	0.062
*P. aeruginosa* RP73	32 (32)	64 (64)	8 (16)	4 (4)	0.31	4 (8)	8 (8)	0.25	4 (4)	1 (1)	0.14
*P. aeruginosa* PAO1	16 (16)	64 (128)	16 (32)	2 (2)	1.03	4 (4)	2 (2)	0.28	4 (4)	0.5 (0.5)	0.25
*P. aeruginosa* PA14	16 (32)	64 (64)	4 (8)	4 (4)	0.31	4 (4)	4 (4)	0.31	4 (4)	1 (1)	0.26
*P. aeruginosa* PA-Na-1	32 (64)	64 (128)	8 (8)	2 (2)	0.28	4 (4)	2 (2)	0.156	4 (4)	0.5 (0.5)	0.132
*S. enterica* ATCC 13076	4 (4)	16 (32)	1 (1)	1 (1)	0.312	1 (1)	1 (1)	0.312	0.5 (0.5)	0.5 (0.5)	0.15
*S. maltophilia* K279	128 (128)	16 (16)	16 (32)	8 (8)	0.62	4 (4)	4 (8)	0.281	4 (4)	2 (2)	0.156
*E. faecalis* ATCC 29212	32 (32)	4 (8)	4 (4)	2 (2)	0.625	0.5 (0.5)	0.5 (0.5)	0.14	0.5 (0.5)	0.5 (0.5)	0.14
*E. faecium* ATCC 6057	8 (8)	4 (4)	0.125 (0.125)	0.5 (0.5)	0.14	0.5 (0.5)	0.125 (0.125)	0.093	0.25 (0.25)	0.06 (0.06)	0.046
*S. aureus* ATCC43300	1 (2)	8 (16)	0.125 (0.125)	0.5 (0.5)	0.187	0.25 (0.25)	0.06 (0.06)	0.257	0.125 (0.125)	0.25 (0.25)	0.156
*C. albicans* ATCC 10231	16 (32)	4 (4)	8 (8)	1 (1)	0.75	2 (4)	0.125 (0.125)	0.156	2 (2)	0.125 (0.0125)	0.156
*C. albicans* 62033	16 (16)	2 (2)	8 (16)	2 (2)	1	4 (4)	2 (2)	0.75	4 (4)	2 (2)	0.75
*C. parapsilosis* 4609	64 (64)	2 (2)	8 (16)	2 (2)	0.51	1 (1)	2 (2)	0.26	1 (1)	2 (2)	0.26

* FIC index, Fractional Inhibitory Concentration index; ^(a)^ Σ FIC = [(MIC CHX + BZK) + 0 RV/MIC CHX] + [(MIC BZK + CHX) + 0 RV/MIC BZK]; ^(b)^ Σ FIC = [(MIC CHX + BZK) + 32 RV/MIC CHX] + [(MIC BZK + CHX) + 32 RV/MIC BZK]; ^(c)^ Σ FIC = [(MIC CHX + BZK) + 64 RV/MIC CHX] + [(MIC BZK + CHX) + 64 RV/MIC BZK]. The numbers outside and within the brackets indicate MIC and MBC or MCF values, respectively, and are expressed in mg/L. The numbers 0, 32 and 64 indicate RV concentrations and are expressed in mg/L.

## Data Availability

Not applicable.

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
