# Peer review of "Resveratrol Reverts Tolerance and Restores Susceptibility to Chlorhexidine and Benzalkonium in Gram-Negative Bacteria, Gram-Positive Bacteria and Yeasts"

_antibiotics, 2022, doi:10.3390/antibiotics11070961_

Round 1
Reviewer 1 Report
The manuscript is well written and organized.
The authors of this research aimed to analyze the susceptibility to the biocides BZK and CHX and if resveratrol was able to restore susceptibility. For that, they resorted to a panel of reference strains and clinical isolates of Gram-negative, Gram-positive bacteria and yeasts.
They have found it to revert tolerance and restore susceptibility in the majority of microorganisms responsible for HAI.
I have no major comment regarding the study, nevertheless, I should note that the graphical representations in Figures 1 and 2 are not, in my opinion, the best for a clear understanding of the results. Moreover, the authors should indicate the threshold between tolerance and susceptibility, allowing the readers to interpret the figures and tables with increasing concentrations of RV and the effect on the determined MIC and MBC (or MFC for the Candida species).
Throughout the manuscript, the usage of italics for species and genes should be ensured.
Author Response
We thank reviewer 1 for her/his positive comments and for considering the manuscript “well written and organized”.
Following reviewer’s 1 suggestion, new files were generated for Figures 1 and 2 to improve the quality and graphics.
The threshold between tolerance and susceptibility has been detailed on lines 349-351 of Materials and Methods section.
The usage of italics for species and genes was ensured throughout the manuscript.
Reviewer 2 Report
In the abstract, the relationship between the “33 analyzed strains” and the “panel of 151 reference strains and clinical isolates” is not clear. Moreover, the fact that RV reduced MIC in “all analyzed strains” but not in certain strains is likely to be confusing, therefore this part of the abstract should be improved for more clarity.
Lines 35-39: this sentence should be appropriately referenced.
Line 313: the concentration range for RV should be justified (in our view, they are relatively high and difficult to achieve in clinical practice, therefore lower concentrations would have been preferrable in order to claim clinical relevance). In the literature RV has been described as having antibacterial and antifungal properties, and particularly for the antifungal effects, MIC values have been reported in the range used in this study (see, e.g. Vestergaard, M., & Ingmer, H. (2019). Antibacterial and antifungal properties of resveratrol. International Journal of Antimicrobial Agents, 53(6), 716-723; Ma, D. S., Tan, L. T. H., Chan, K. G., Yap, W. H., Pusparajah, P., Chuah, L. H., ... & Goh, B. H. (2018). Resveratrol—potential antibacterial agent against foodborne pathogens. Frontiers in Pharmacology, 9, 102 etc) These aspects should also be discussed in the introduction and the discussion section.
Line 360: ANOVA is commonly used together with a post-hoc test. If such a test was used, it should be clarified which one.
Line 162: whereas the correlation coefficient may be useful as a point estimate of correlation, it is essential that a visual representation is provided to allow the reader a glimpse into the correlation (a scatter plot).
Table 1: it should be clarified what the numbers outside brackets and what the numbers within brackets correspond to (now the reader has to guess).
The Results section is limited to results. There is no critical discussion of the results in the context of the current state of knowledge. Either a Discussion(s) section should be added or the current section should be supplemented with a few paragraphs of discussions.
The fact that tables are provided may be useful to understand individual observations, but we would be interested in a discussion of the trends and there consistency: is there a consistent effect seen to all RV concentrations? Or is the effect concentration dependent? Appropriate plots would be useful for this aspect.
Author Response
We thank reviewer for her/his detailed comments.
Query: In the abstract, the relationship between the “33 analyzed strains” and the “panel of 151 reference strains and clinical isolates” is not clear. Moreover, the fact that RV reduced MIC in “all analyzed strains” but not in certain strains is likely to be confusing, therefore this part of the abstract should be improved for more clarity.
Response: The abstract was rewritten to avoid confusion among the number of strains.
Query: Lines 35-39: this sentence should be appropriately referenced.
Response: Four new citations were quoted to appropriately reference sentence on line 35-39.
Query: Line 313: the concentration range for RV should be justified (in our view, they are relatively high and difficult to achieve in clinical practice, therefore lower concentrations would have been preferrable in order to claim clinical relevance). In the literature RV has been described as having antibacterial and antifungal properties, and particularly for the antifungal effects, MIC values have been reported in the range used in this study (see, e.g. Vestergaard, M., & Ingmer, H. (2019). Antibacterial and antifungal properties of resveratrol. International Journal of Antimicrobial Agents, 53(6), 716-723; Ma, D. S., Tan, L. T. H., Chan, K. G., Yap, W. H., Pusparajah, P., Chuah, L. H., ... & Goh, B. H. (2018). Resveratrol—potential antibacterial agent against foodborne pathogens. Frontiers in Pharmacology, 9, 102 etc) These aspects should also be discussed in the introduction and the discussion section.
Response: That “RV showed no antimicrobial activity against all gram-negative bacteria, gram-positive bacteria and yeasts included in the study with MIC values >1,024 mg/L (Table S1)” has been detailed on lines 141-144 of revised manuscript and RV MIC values include in Table S1 for all 151 strains. The antimicrobial and antifungal properties of RV reviewed in the references kindly suggested by reviewer 2 were quoted in the introduction section (lines 58-59) and the discrepancies with our data discussed on lines 144-148.
Query: Line 360: ANOVA is commonly used together with a post-hoc test. If such a test was used, it should be clarified which one.
Response: The information that ” The significance of differences was evaluated using one-way ANOVA followed by Bonferroni’s comparison post-hoc tests” has now been provided on lines 377-378 of revised manuscript.
Query: Line 162: whereas the correlation coefficient may be useful as a point estimate of correlation, it is essential that a visual representation is provided to allow the reader a glimpse into the correlation (a scatter plot).
Response: Scatter plots of the effects of RV or CCCP on CHX MICs and RV or CCCP on BZK MICs were provided as supplementary Figure S1 and Figure S2, respectively, and cited in the text of revised manuscript on line 158 and line 167, respectively.
Query: Table 1: it should be clarified what the numbers outside brackets and what the numbers within brackets correspond to (now the reader has to guess).
Response: the information that “the numbers outside and within the brackets indicate MIC and MBC or MCF values, respectively” has been introduced in the footnotes of Tables 1-4 as suggested.
Query: The Results section is limited to results. There is no critical discussion of the results in the context of the current state of knowledge. Either a Discussion(s) section should be added or the current section should be supplemented with a few paragraphs of discussions.
Response: As per reviewer’s 2 suggestion, the following paragraphs of discussion were added to the result and discussion section at lines 93-96, 128-133, 140-148, and 295-298.
Query: The fact that tables are provided may be useful to understand individual observations, but we would be interested in a discussion of the trends and there consistency: is there a consistent effect seen to all RV concentrations? Or is the effect concentration dependent? Appropriate plots would be useful for this aspect.
Response: RV effects on CHX and BZK MICs is dose-dependent as explained on lines 152-155 and 162-165, respectively, and described in more details by scatter plots provided as FigureS1 and Figure S2, respectively.
Reviewer 3 Report
Point 1- Introduction
- Very brief, and with few bibliographical references. A more complete bibliographical research on the subject should be done. Namely of similar studies already published on these antiseptics and microorganisms under study.
- Make a summary table of what has already been published.
You must explain the importance, innovation, novelty of this study in relation to what has already been published on this subject.
In section 2 - Discussion and results
- In the text some microorganisms are identified with ATCC numbers and others are not. This identification should be uniform in the text ( methods).
- Increase the quality of the graphics of figure 1. The legends are not well visible.
- Insert the data ( MIC...) obtained for the positive and negative controls
- Clarify text line 144 and 145.
- Complete and compare your values with others in the literature with references to other similar works.
4- Materials and methods
- Clarify the origin of the strains, collection conditions, laboratory, storage and identification.
Author Response
We thank reviewer for her/his comments.
Queries:
Point 1- Introduction
- - Very brief, and with few bibliographical references. A more complete bibliographical research on the subject should be done. Namely of similar studies already published on these antiseptics and microorganisms under study.
- - Make a summary table of what has already been published.
- You must explain the importance, innovation, novelty of this study in relation to what has already been published on this subject.
Responses:
- Nine new references were quoted in the Introduction section.
- We believe that a summary table of what has been published on the subject is appropriate for either systematic or descriptive review article, but not for an original article.
- The importance, innovation, and novelty of the study in relation to what has been published on this subject has been explained on lines 53-65.
Queries:
In section 2 - Discussion and results
a) - In the text some microorganisms are identified with ATCC numbers and others are not. This identification should be uniform in the text (methods).
b) - Increase the quality of the graphics of figure 1. The legends are not well visible.
c)- Insert the data ( MIC...) obtained for the positive and negative controls
d)- Clarify text line 144 and 145.
e)- Complete and compare your values with others in the literature with references to other similar works.
Responses:
a)- The information that “A collection of 132 Gram-negative bacteria, 9 Gram-positive bacteria and 10 Candida spp. strains was analyzed in the study (Table S1). The collection included either reference strains, which were identified with their ATCC number, or clinical isolates, which were identified with their original number (Table S1)” was provided on lines 313-317 of the Materials and Methods section.
b)- New files were generated for Figures 1 and 2 to improve the quality and graphics.
c)- The positive controls correspond to non-treated bacteria or yeasts, while the negative controls to wells containing only CAHMB or MOPS media, respectively. This has been explained in the Materials and Methods section paragraph on lines 334-344.
d) That “RV showed no antimicrobial activity against all gram-negative bacteria, gram-positive bacteria and yeasts included in the study with MIC values >1,024 mg/L (Table S1)” has been detailed on lines 140-141 of revised manuscript and RV MIC values include in Table S1 for all 151 strains. The antimicrobial and antifungal properties of RV reviewed in the references kindly suggested by reviewer 2 were quoted in the introduction section (lines 58-59) and the discrepancies with our data discussed on lines 144-148.
e) The following paragraphs of discussion were added to the result and discussion section at lines 93-96, 128-133, 140-148, and 295-298 to compare data shown herein with previous published data.
Query:
4- Materials and methods
- Clarify the origin of the strains, collection conditions, laboratory, storage and identification.
Response:
The origin and characteristics of all strains include in the study were described in the references listed in Table S1. The above information has been provided also on lines 316-317 of Materials and Methods section.
Reviewer 4 Report
General comment
This manuscript demonstrates that resveratrol shows a synergic or additive microbicidal effect combined with two widely used biocides (CHX and BZ). The bactericidal effect has been proven against a large panel of Gram-negative, Gram-positive bacteria and Candida yeasts. Although the effect seems to be general, the potency depends on the nature of the bacterial strains. Some strains are more tolerant, and the combined effect would need higher doses of the three agents. The study identifies the most sensitive strains. The enhancing biocide effect seems to be mediated by inhibition of the efflux pumps (EPs) expression, although this point has been described previously level of EPs. These efflux pumps are main responsible of the tolerance developed by some nosocomial dangerous strains to cationic biocides CHX and BZK by extruding them through Eps.
These results have a potential application for preparing disinfectant/antiseptic formulations containing the three components against tolerant Gram-negative, Gram-positive bacteria and yeasts. However, the degree of the effect depends on the strain according to the MIC, MBC (and MFC in the case of Candida spp.) values reported in a series of Tables including in the manuscript and a Table S1 provided as supplementary material.
The manuscript is absolutely suitable for publication once some particular details be addressed in order to improve the presentation quality.
Line 86: “values ranging from 4 to 64 mg/L and 4 to 64 mg/L, respectively, and were considered”. Re-write using 4-64 mg/L for both agents to avoid repetition.
Figure 1 and 2 should be increased in quality to facilitate the reading of the labels and axes scale.
Line 144: It is stated that the concentration of RV tested somewhere is >1,024 mg/L. At methods, line 341, the range of resveratrol concentrations is 32–256 mg/L (line 341). Please, check this point and clarify. Is the synergic effect of RV greater using concentrations higher than 256 mg/L in the case of the most resistant bacterial strains?
Data using the carbonyl cyanide m-chlorophenylhydrazone protonophore (CCCP) as efflux pump inhibitor are complementary, as the paper is devoted to RV effect. Thus, Table 1 could be placed as supplementary paper. Other Tables should indicate at the legend at the upper row are the concentrations of the agent studied.
Line 281: a synergic or additive effect in 20 and 1 strains (Table 4). Please, write 21 strains. In general, I recommend that the expression of the number of strains would be unified, using always figures (see for instance One-hundred-thirty-two Gram-negative bacteria at line 296). .
Author Response
We thank reviewer for her/his comments and for considering the manuscript “absolutely suitable for publication once some particular details be addressed in order to improve the presentation quality”.
Query: Line 86: “values ranging from 4 to 64 mg/L and 4 to 64 mg/L, respectively, and were considered”. Re-write using 4-64 mg/L for both agents to avoid repetition.
Response: This has been corrected as suggested.
Query: Figure 1 and 2 should be increased in quality to facilitate the reading of the labels and axes scale.
Response: New files were generated for Figures 1 and 2 to improve the quality and graphics.
Query: Line 144: It is stated that the concentration of RV tested somewhere is >1,024 mg/L. At methods, line 341, the range of resveratrol concentrations is 32–256 mg/L (line 341). Please, check this point and clarify. Is the synergic effect of RV greater using concentrations higher than 256 mg/L in the case of the most resistant bacterial strains?
Response: : RV effects on CHX and BZK MICs is dose-dependent as explained on lines 152-155 and 162-165, respectively, and described in more details by scatter plots provides as FigureS1 and Figure S2, respectively. To address your query and clarify this point, the following sentence was added on line 149-151 of results and discussion section: "The objectives of our experiments were to identify which were RV concentrations able to revert tolerance and restore susceptibility to CHX and BZK biocides."
Query: Data using the carbonyl cyanide m-chlorophenylhydrazone protonophore (CCCP) as efflux pump inhibitor are complementary, as the paper is devoted to RV effect. Thus, Table 1 could be placed as supplementary paper. Other Tables should indicate at the legend at the upper row are the concentrations of the agent studied.
Response: Because data shown in Table 1 are important to follow comparison analysis with data shown in Tables 2 and 3 and several paragraphs of discussion in results and discussion section, we would prefer to leave Table 1 in the main text of the manuscript. As per reviewer's request, it has been indicated in the legends of Tables 1-4 that the numbers at the upper row of the tables "indicate concentrations of the agents and are expressed in mg/L ".
Query: Line 281: a synergic or additive effect in 20 and 1 strains (Table 4). Please, write 21 strains. In general, I recommend that the expression of the number of strains would be unified, using always figures (see for instance One-hundred-thirty-two Gram-negative bacteria at line 296).
Response: This has been corrected as suggested.
Round 2
Reviewer 2 Report
The manuscript has definitely improved and in our view it can be published.
Reviewer 3 Report
Accept in present form.